# Untargeted Lipidomics after D_2_O Administration Reveals the Turnover Rate of Individual Lipids in Various Organs of Living Organisms

**DOI:** 10.3390/ijms241411725

**Published:** 2023-07-21

**Authors:** Yury Kostyukevich, Elena Stekolshikova, Anna Levashova, Anna Kovalenko, Anna Vishnevskaya, Anton Bashilov, Albert Kireev, Boris Tupertsev, Lidiia Rumiantseva, Philipp Khaitovich, Sergey Osipenko, Eugene Nikolaev

**Affiliations:** 1Skolkovo Institute of Science and Technology, Bolshoy Boulevard 30, Bld. 1, 121205 Moscow, Russia; e.stekolschikova@skoltech.ru (E.S.); a.levashova@skoltech.ru (A.L.); a.kovalenko@skoltech.ru (A.K.); ai.vish@yandex.ru (A.V.); a.bashilov@skoltech.ru (A.B.); a.kireev@skoltech.ru (A.K.); b.tupertsev@skoltech.ru (B.T.); lidiia.rumiantseva@skoltech.ru (L.R.); p.khaitovich@skoltech.ru (P.K.); sergey.osipenko@skoltech.ru (S.O.); e.nikolaev@skoltech.ru (E.N.); 2Scientific Center of Biomedical Technologies of the Federal Medical and Biological Agency, Krasnogorsky District, Village Light Mountains, Bld. 1, 143442 Moscow, Russia

**Keywords:** D_2_O, metabolism, lipids, LC-MS/MS

## Abstract

The administration of low doses of D_2_O to living organisms was used for decades for the investigation of metabolic pathways and for the measurement of the turnover rate for specific compounds. Usually, the investigation of the deuterium uptake in lipids is performed by measuring the deuteration level of the palmitic acid residue using GC-MS instruments, and to our knowledge, the application of the modern untargeted LC-MS/MS lipidomics approaches was only reported a few times. Here, we investigated the deuterium uptake for >500 lipids for 13 organs and body liquids of mice (brain, lung, heart, liver, kidney, spleen, plasma, urine, etc.) after 4 days of 100% D_2_O administration. The maximum deuteration level was observed in the liver, plasma, and lung, while in the brain and heart, the deuteration level was lower. Using MS/MS, we demonstrated the incorporation of deuterium in palmitic and stearic fragments in lipids (PC, PE, TAG, PG, etc.) but not in the corresponding free forms. Our results were analyzed based on the metabolic pathways of lipids.

## 1. Introduction

Changes in lipid metabolism regulation play an important role in the development of many diseases. Obesity leads not only to lipid accumulation in adipose tissue but also to increased lipid storage in ectopic tissues, such as skeletal muscle, liver, kidney, etc. [1]. Lipid accumulation in the heart may result in cardiac dysfunction and heart failure. Lipid accumulation in the kidneys causes the alteration of kidney function [2], with diabetic nephropathy being the most common cause of progressive kidney disease. Currently, the most common chronic liver disease in the world is non-alcoholic fatty liver disease (NAFLD), affecting 20–30% of the adult population. It is associated with hepatic insulin resistance, and it is a risk factor for the development of type 2 diabetes [2,3]. The nervous system and many organs need a continuous replacement of the components of membranes to continue functioning well [4].

The investigation of lipid metabolism and lipid turnover in the various tissues and organs of living organisms dates back to the classical work of R. Schoenheimer in the 1930s [5,6,7,8]. He developed an approach of the administration of heavy water (D_2_O) to the living organism followed by the measuring of the deuterium content in the different organs and classes of biomolecules. Later this approach was actively developed, including the use of both stable and radioactive isotopes. M.E. Smith reported the use of [^14^C]glucose to measure the turnover rate of many lipids and proteins in the myelin of adult rats [9]. He measured the half-lives of inositol phosphatide (18 days), lecithin (41 days), serine phosphatide (4 months), sphingomyelin (8 months), ethanolamine phosphatide (6 months), cerebroside (1 year), and cholesterol (8 months). Similar results were reported by other authors [10,11].

Many researchers reported the application of the administration of D_2_O coupled with mass spectrometry for the investigation of lipid metabolism [3,4,10,11,12,13,14]. Allister et al. investigated the impaired triglyceride storage in adipose tissue of insulin-resistant humans. Subjects consumed D_2_O for 4 weeks before adipose tissue biopsy to achieve and maintain a total body D_2_O water enrichment of 2%. Isotope enrichments were measured by GC-MS analysis. The authors demonstrated that the synthesis of fatty acids was 63% lower in insulin-resistant subjects as compared with insulin-sensitive subjects [13]. Ando et al. measured the turnover of myelin lipids in the aging brain and demonstrated that the incorporation of cholesterol was fast in infant brains and fixed at a slower rate in adult brains [4]. The incorporation rate of phosphatidylcholine and phosphatidylethanolamine was almost constant. Castro-Perez et al. used D_2_O administration coupled with GC-MS and LS-MS/MS to measure the changes in cholesterol synthesis for a high-carbohydrate and high-fat diet [3]. Fu et al. reported the use of high-resolution Orbitrap for the resolving of ^13^C and ^2^H isotopes of deuterium-labeled fatty acids [15]. Application of the isotope ratio mass spectrometry was also reported [16,17].

Recently, our research group worked on the application of an isotope exchange approach (both H/D and ^16^O/^18^O) for targeted [18,19,20,21,22] and untargeted [23,24,25,26,27,28,29,30,31,32,33] screening of compounds in complex natural and biological matrixes. We have developed an analytical platform for the enumeration of the number of isotope exchanges for all compounds in an LC-MS/MS run. Application of those approaches to the analysis of in vivo labeled lipids may bring new insights for the understanding of lipid metabolism.

Here, we describe the use of the non-targeted LC-MS/MS lipidomics workflow for the simultaneous determination of the turnover rate of different lipids in different organs of mice.

## 2. Results

Briefly, our experiment can be described as follows. Pure deuterium oxide (D_2_O) was administrated to two mice for 7 days. No other liquids were given. Despite the fact that living organisms cannot tolerate 100% D_2_O for a long time, we still decided to work with 100% D_2_O in order to achieve the maximum possible deuteration level. Mice were killed at the beginning of the dramatic decrease in well-being after 1 week, so they did not suffer. The concentration of deuterium in plasma was measured by IR spectroscopy, and after 7 days, it was approximately 60%. Blood and several organs were collected, and lipids were extracted and analyzed using HPLC-MS/MS. We identified more than 650 lipids in the negative ion mode and assigned potential candidates to more than 1000 lipids in the positive ion mode. Details are given in Section 4. The tables of lipids can be found in the Appendix A.

We clearly observed the deuterium incorporation in all detected lipids. In Figure 1, we present the deuterium distribution for LPC (lysophosphatidylcholine) 20:4, LPC 16:0, and PC (phosphatidylcholine) 16:0_20:2 for the brain and liver. The accurate mass difference corresponding to the H/D exchange is 1.006277. Unfortunately, our experiments were performed with the maximum resolving power of 140,000 (Orbitrap QExactive mass spectrometer), so we were not able to resolve natural [^13^C] isotopic peaks from the peaks corresponding to the incorporation of deuterium. However, the remarkable changes in the shape of the isotopic distribution (the appearance of many new peaks shifted to the right) and close values of mass differences to that corresponding to the H/D exchange prove considerable deuterium uptake. New peaks corresponding to the deuterium exchange are marked with a red rectangle (Figure 1).

We can see that for each lipid, the deuterium incorporation is considerably higher in the liver than in the brain. Also, the deuterium incorporation in LPC 16:0 and PC16:0_20:2 is higher than in LPC 20:4. This is expected because the deuterium preferably incorporates in the palmitic acid (16:0) via biochemical synthesis. As a consequence, we can expect higher deuterium incorporation for lipids containing palmitic (16:0) and stearic (18:0) acids. A low rate of deuterium incorporation is expected for non-saturated acids, which also can be synthesized in organisms; however, they mainly enter the body via consuming foods.

In Figure 2, we present the obtained deuterium distribution for 13 lipids for several selected organs, plasma, and urine. Here, we selected only those lipids for which we observed high-quality spectra for all organs. We can see that the shape of the isotopic distribution and, as a consequence, the number of deuterium uptakes is different for different organs. Generally, the highest deuterium uptake was observed in plasma, lung, and liver. The deuterium uptake for the brain, heart, fat, kidney, spleen, erythrocytes, and lymphocytes was considerably lower. We can also see the lipids containing saturated acids (16:0 and 18:0) produce an elongated to the right deuterium distribution.

If we take a closer look at the deuterium distribution of the selected lipids in the lung, we will see that the lipid PG (Phosphatidylglycerol) 16:0_18:2 demonstrated a considerable deuterium uptake compared to all other lipids. Moreover, the deuterium uptake of this lipid is higher in the lung than in the liver. Such results can indicate the fast synthesis of this lipid in the lung. This is in agreement with the fact that phosphatidylglycerols are secreted by the alveolar wall surface and that pulmonary surfactant consists of phosphatidylcholine (primarily dipalmitoylphosphatidylcholine), forming 40% of surfactant and phosphatidylglycerols, forming 10% of surfactant.

In order to get a general estimation of the deuterium uptake rate and compare it for different organs, we calculated the average shift of the isotopic distribution for each lipid. Precisely, the following value was calculated:D_shift_ = sum (m_n_ × I_n_)/sum (I_n_) − m_0_,
where m_n_ and I_n_ are the *m*/*z* and intensity of n-peak in the isotopic envelope.

The value of a D_shift_ corresponds to the rate of the deuterium uptake for each lipid. It is high when the rate of uptake is high. We calculated the D_shift_ for >150 lipids, for which we obtained reliable identifications and high-quality isotope distribution spectra. The table is given in the Appendix A. In Figure 3, we demonstrate the average values (density distribution) of the D_shift_ for all organs and for several major classes of lipids: LPC, LPE, PG, PC, and PE. Each curve represents the distribution of the values of D_shift_ for all lipids in the corresponding class. Figure 3 demonstrates the difference in the rate of deuterium uptake for all organs and classes of lipids. We can see that, generally, the rate of deuterium uptake for PC, PE, LPC, and LPE is almost the same in the selected organ. Generally, the rate of uptake in PC and LPC is a little bit higher than in PE and PLE, correspondingly. The rate of uptake decreases in the following series: liver–plasma–lymphocytes–spleen–lung–kidney–fat (adipose tissue)–heart-brain–erythrocytes. The number of detected lipids in cells and urine is not considerable. For PG lipids, the highest rate of deuterium uptake was observed in the lung, liver, kidney, spleen, and urine. Our results allow us to estimate the rate of synthesis of lipids in various organs.

In order to prove that deuterium is incorporated in lipids, preferably in saturated fatty acids, we performed the following experiment. For the selected lipid, we obtained fragmentation spectra for all peaks corresponding to the different numbers of incorporated deuterium. Fragmentation pathways for lipids of different classes are shown in Figure 4. Fragmentation of TAG (Triglyceride) in the positive mode (in the form of [M + NH_4_]^+^) leads to the loss of each fatty acid residue. We observed fragments without corresponding residues. In the fragmentation spectrum of PE in negative mode (in the form of deprotonated ion), we observed ions corresponding to the fatty acid residues and ions corresponding to the loss of fatty acid from specific sites.

Our results are presented in Figure 5. Fragmentation of TG 16:0_18:1_18:1 leads to the loss of each fatty acid residue. We can see that when the 18:1 fragment is lost, the number of deuterium in the fragment ion seems to not decrease, while when the 16:0 fragment is lost, the shape of the deuterium distribution changes considerably. However, the fact that, even when the 16:0 fragment is lost, we still observe the peak with the maximum number of deuteriums means that deuterium can also incorporate in the glycerol or in 18:1.

Fragmentation of PE 18:0_20:4 results in the three fragment ions, two of them are fatty acid residues 18:0 (*m*/*z* = 283) and 20:4 (*m*/*z* = 303), and ion *m*/*z* = 480 corresponds to the loss of a fragment of arachidonic acid (see Figure 4). We can clearly see that the deuterium primarily incorporates in the stearic acid (18:0) and not in arachidonic (20:4). The loss of the arachidonic acid also produces ions with the maximum number of deuteriums.

When fragmenting PC lipids in positive ion mode, we never observed any incorporation of deuterium in 184+ ions (see Figure 6). That means that all deuterium was incorporated in the fatty acids and not in the choline.

Deuterium distribution for other lipids, fragmentation spectra, and other initial experimental files are available in the Appendix A.

## 3. Discussion

The inclusion of the isotopic labels is primarily associated with the active biosynthesis of fatty acids (Figure 7). The possible sites of the deuterium from D_2_O incorporation in the fatty acid are shown in red. The biosynthesis of fatty acids in mammals occurs in the cytosol and originates from the junction of malonyl-CoA and Acetyl-CoA residues. These fragments are formed in the cell from the Krebs cycle occurring in the mitochondria. In the Krebs cycle, there is a hydroxylation reaction catalyzed by the enzyme fumarase. As a result of this reaction, two deuteriums from heavy water can be introduced into the composition of the resulting malate. Deuterium in the hydroxyl group is mobile, so we assume that deuterium bound to the third carbon is primarily retained.

From the Krebs cycle, intermediates for the biosynthesis of fatty acids exit the mitochondria at the citrate level, which is then converted to acetyl-CoA and malonyl-CoA. As a result of transformations in acetyl and malonyl residues, deuterium remains bound to the second carbon atom. For each new cycle of the fatty acid synthase, two carbon atoms are attached to the fatty acid molecule (the third atom in the composition of the malonyl residue is decarboxylated as a result of the initial stages of the fatty acid synthase). As a result of such synthesis, the inclusion of eight deuterium isotopes bound to every second carbon atom in fatty acids 16:0 should be expected.

It is evident that the accurate understanding of the lipid biosynthesis pathways in different organs using deuterium exchange will also require measuring the deuterium incorporation in the intermediates of the Krebs cycle, NADH, and other compounds.

It is also important to discuss the possible incorporation of the deuterium into the residues of glycerin, ethanolamine, and choline. The biosynthesis of these lipid fragments is depicted in Figure 8.

Despite the fact that at the stage of serine formation preceding the biosynthesis of choline and ethanolamine, a hydration reaction occurs, deuterium, which can come from D_2_O, does not bind to carbon, therefore it is labile and is not visible in the analysis of HPLC-MS.

Further work on the improving of HPLC separation of lipids and use of higher resolving power, as well as the optimization of the D_2_O feeding protocol, are required for obtaining results of better quality and further understanding of the metabolic pathways and turnover rates of lipids in different tissues.

## 4. Materials and Methods

**Mouse handling**. The six male C57BL/6J mice used in this study had an average age of 8 weeks, with an average weight of 20 ± 2.0 g. The mice were kept in the RairIsoSystemmicroisolator system. Animals were satisfied with conventional category. Regular rodent food (standard granular compound feed for laboratory animals (extruded) PK-120 GOST R 51849-2001 R.5) was provided. For the labeling experiments, deuterated water (100%) for experimental animals and filtered tap water for control animals were provided ad libitum. The climate was maintained at a room temperature of 22 °C (+/−2), room humidity at 60–70%, and a 12/12 light/dark cycle regimen. All experiments were carried out in accordance with the ethical principles and regulations recommended by the European Convention for the Protection of Vertebrate Animals used for Experiments.

After 7 days, mice were sacrificed and immediately decapitated, and the full blood samples were collected directly from eye into EDTA tubes. For plasma preparation, tubes were centrifuged for 5 min at 3000× *g*, and plasma was collected within 1 h after blood collection. Samples were frozen at −80 °C until lipid extraction.

**Organs and tissues** were dissected from fresh cadavers after all blood was drained (no perfusion was performed), resulting in the following samples: heart, liver, brain, kidney, lung, and spleen adipose tissue (from the abdominal region). Sterile urine was collected with a sterile needle from the bladder. Lymphocyte fraction was obtained in a ficoll gradient according to the standard protocol. These organs, cells, and tissues were put into 2 mL microcentrifuge tubes and immediately frozen on dry ice at −80 °C until homogenization.

**Sample preparation**. For liquids (plasma, blood, urine), the following protocol was used:
A total of 300 mL of cold methanol was added to 40 mL of aliquots of the sample and vigorously shaken on a shaker for 1 min;Then, 1 mL of cold MTBE was added, and the mixture was treated with ultrasound for 10 min and incubated for 40 min at 4 °C with stirring;A total of 250 mL of water was added to the extract to separate the phases. The extract was shaken for 1 min at 4 °C, then centrifuged for 10 min at 13,000 rpm at 4 °C;An aliquot of 1000 µL of the upper layer containing nonpolar components was collected in a separate vial;A total of 400 µL of buffer (MeOH:MTBE:H_2_O = 3:10:2.5) was added to the lower phase for repeated extraction;The sample was shaken and centrifuged for 10 min at 13,000 rpm at 4 °C. The upper fractions were combined (a total of 1300 µL) and evaporated dry in a vacuum concentrator at room temperature;The dry residue was re-dissolved in 200 mL of a mixture of acetonitrile: isopropanol cooled to 0 °C (7:3 (*v*/*v*));The sample was shaken for 10 min, kept in an ice-cooled ultrasonic bath for 10 min, and centrifuged for 5 min at 13,000 rpm;Before the HPLC MS analysis, the samples were diluted 1:5 and 1:2 with a mixture of acetonitrile: isopropanol (7:3 (*v*/*v*)) for measurements in the registration mode of positively and negatively charged ions, respectively.

Cells were suspended in 100 µL of MeOH and sonicated for 10 min prior to applying the above protocol.

Tissues were homogenized in 4:1 MeOH (to mass of tissue) prior to applying the above protocol.

**LC-MS/MS analysis**. The mice tissue lipidome profiling was performed on a Waters Acquity UPLC system (Waters, Manchester, UK) coupled with Q-Exactive Orbitrap mass spectrometer (Thermo Fisher, CA, USA) equipped with a heated electrospray ionization (HESI) probe. The lipid extracts were separated on a Reversed Phase Bridged Ethyl Hybrid (BEH) C8 column (100 mm × 2.1 mm, 1.7 µm, Waters) coupled with a Vanguard pre-column with the same solid phase. A binary solvent system used for the chromatographic separation consisted of Buffer A (water containing 10 mM ammonium acetate, 0.1% formic acid) and Buffer B (acetonitrile:isopropanol (7:3 (*v:v*)) containing 10 mM ammonium acetate, 0.1% formic acid). The gradient for separation was programmed as follows: 1 min 55% B, 3 min linear gradient from 55% to 80% B, 8 min linear gradient from 80% B to 85% B, and 3 min linear gradient from 85% B to 100% B. After 4.5 min washing with 100% B, the column was re-equilibrated with 55% B for 4.5 min. The flow rate was kept at 400 µL/min during the whole 24 min run, and the column temperature was set at 60 °C. For negative polarity, formic acid was substituted with acetic acid.

HESI source tune parameters for ionization were set as follows: 320 °C; aux gas heater temperature: 350 °C; capillary voltage: 4.5 kV (−3.5 kV); S-lens RF level: 60; sheath gas flow rate (N_2_): 45 arbitrary units (a.u.); auxiliary gas flow rate (N_2_): 20 a.u., sweep gas flow rate (N_2_): 4 a.u. and kept identical for both polarities. Data acquisition was performed in data-dependent mode in positive and negative polarities separately. For the full scan events, operating parameters were set as listed: resolution: 140,000 at *m*/*z* 200, automatic gain control (AGC target): 5 × 10^5^, maximum injection time (IT): 50 ms, scan range: 100 to 1500 Da (positive polarity) and 200 to 1800 Da (negative polarity). For DDA mode: topN:10, resolution 17,500 at *m*/*z* 200, AGC: 2 × 10^4^, IT: 100 ms, mass isolation window: 1.2 Da, retention time window width: expected time ± 1 min, stepped normalized collision energy: 15, 25, 30%, dynamic exclusion 12 s, inclusion: on, customize tolerances: 10 ppm. The spectra were recorded in the profile mode. Inclusion lists were prepared base on in-house lipid database. Injection volume of 3 μL in both positive and negative ionization modes was used.

**Data processing**. Data processing and lipid identification was performed by freely available MS-DIAL (RIKEN) software V.4.9. Thermo. Raw data format was converted to Analysis Base File (.abf) to import the data into MS-DIAL. Mass accuracy for MS and MS/MS peak centroiding was set to 0.01 and 0.05 Da, respectively. Retention time was not taken into consideration for the calculation of the total score. Accurate mass tolerances were set to 0.01 Da and 0.05 Da for MS^1^ and MS^2^ ions, accordingly. Lipids were annotated according to their accurate precursor and fragment characteristic ions. All identified lipids were manually curated, and then expected lipid elution patterns were considered for further confirmation. Namely, retention time should increase for the series of homologues and decrease with addition of double bonds.

Analysis of the deuterium distribution was performed semi-manual. For each identified lipid, we have extracted spectra at the corresponding retention time in the vicinity of the corresponding *m*/*z* and visually analyzed the shape of the isotopic distribution and the mass difference between peaks. The main problem affecting the analysis was the short elution time and the overlapping of deuterium peaks with ^13^C peaks and peaks corresponding to other lipids.

## 5. Conclusions

We have investigated the deuterium uptake for >500 lipids for 13 organs of mice after the D_2_O administration using the conventional untargeted lipidomics workflow. Using MS/MS, the incorporation of the deuterium into the saturated fatty acid residues was demonstrated. The maximum deuteration level of lipids was observed in the liver, plasma, and lung, while in the brain and heart, the deuteration level was low. We observed that the deuterium was not incorporated into the choline. It was demonstrated that the combination of modern untargeted lipidomics and in vivo deuterium exchange can help to understand the biosynthesis pathways and measure the turnover rate for all lipids simultaneously.

## Figures and Tables

**Figure 1 ijms-24-11725-f001:**
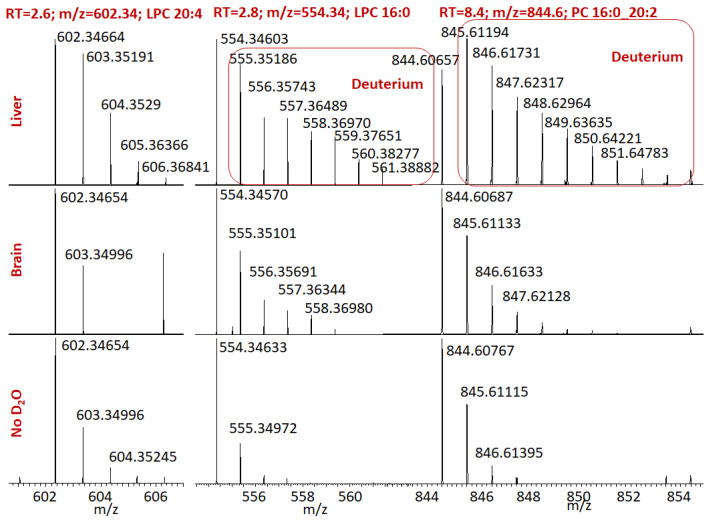
Isotopic distribution for selected lipids in brain and liver after D_2_O administration and normal isotopic distribution of those lipids. RT is the retention time in minutes. Negative ESI mode.

**Figure 2 ijms-24-11725-f002:**
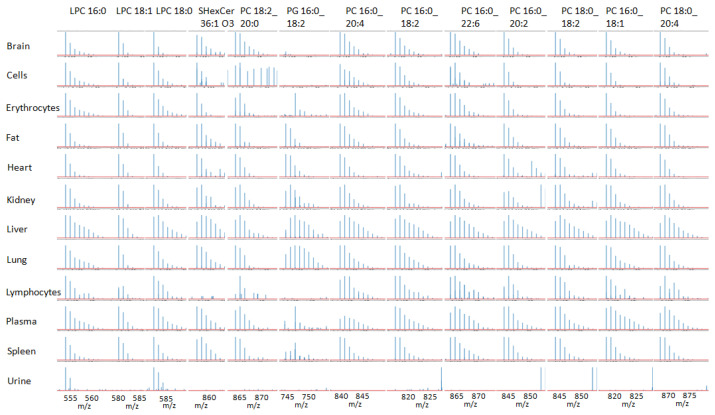
Deuterium distribution for 13 lipids in different organs. Negative ESI mode.

**Figure 3 ijms-24-11725-f003:**
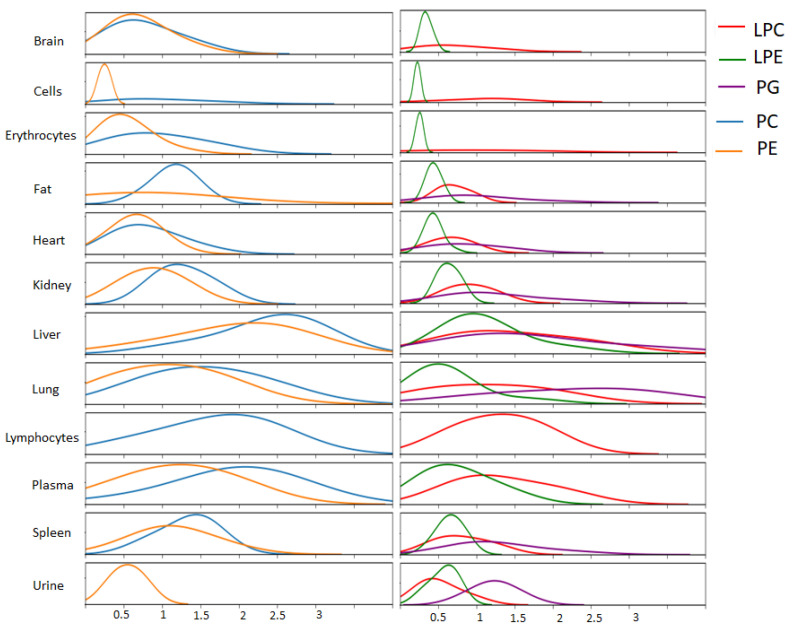
The density distribution of the level of the deuterium uptake for different classes of lipids in all organs.

**Figure 4 ijms-24-11725-f004:**
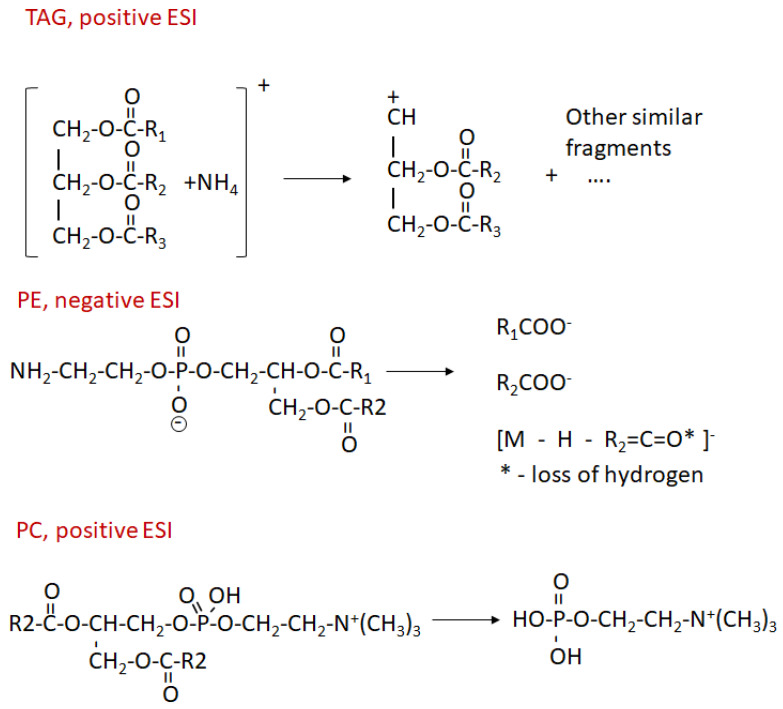
Fragmentation pathways for TAG, PE, and PC lipids. For TAG … means all similar fragments produced by the loss of other residues.

**Figure 5 ijms-24-11725-f005:**
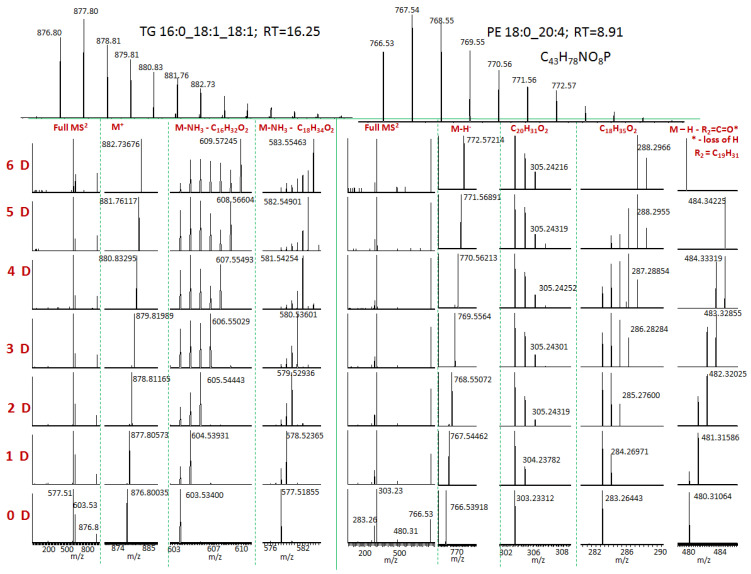
Fragmentation of peaks corresponding to the different numbers of incorporated deuterium. TG in positive ESI mode and PE in negative ESI mode.

**Figure 6 ijms-24-11725-f006:**
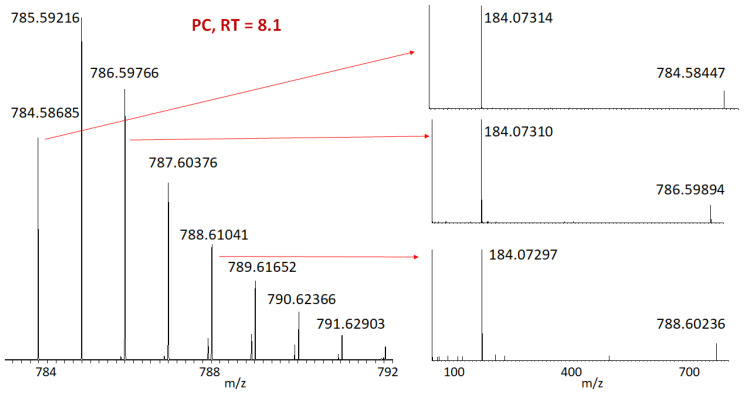
Fragmentation of peaks corresponding to the different numbers of incorporated deuterium. Positive mode. RT is the retention time in minutes.

**Figure 7 ijms-24-11725-f007:**
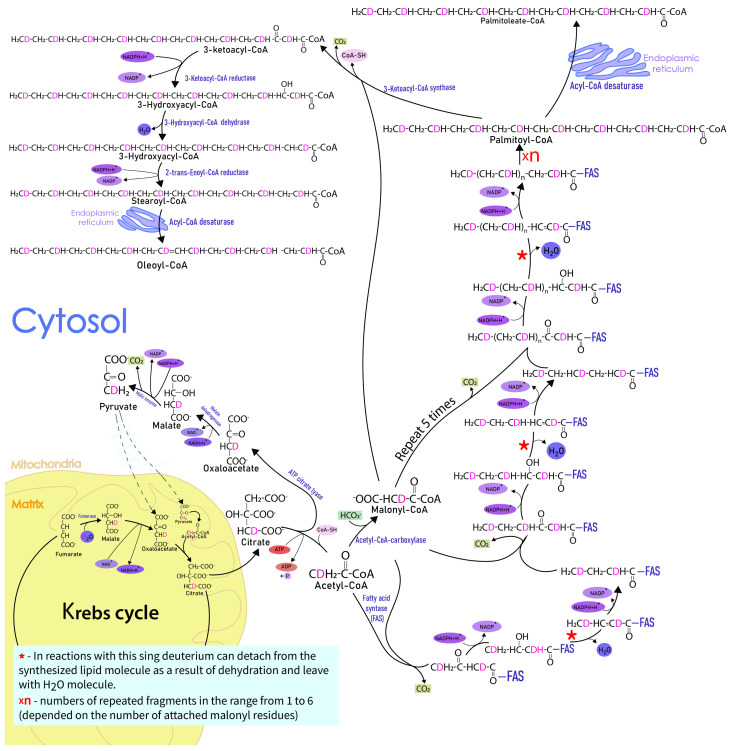
Biosynthesis of fatty acids (palmitic, palmitoleic, stearic, and oleic) in mice cells with the possible sites of deuterium incorporation.

**Figure 8 ijms-24-11725-f008:**
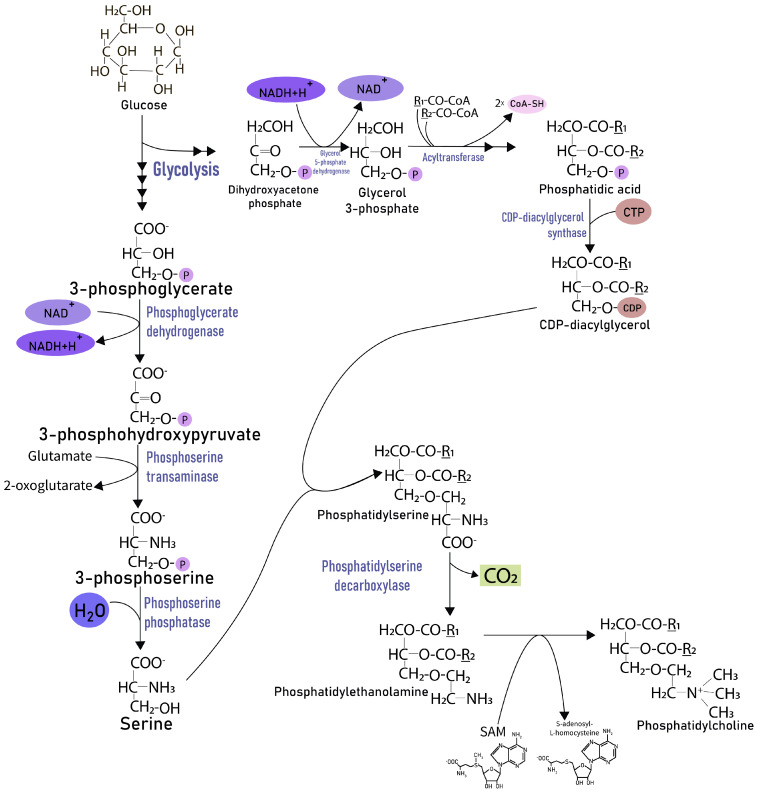
Synthesis of choline.

## Data Availability

All data can be requested from authors.

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
