# Peer review of "Untargeted Lipidomics after D2O Administration Reveals the Turnover Rate of Individual Lipids in Various Organs of Living Organisms"

_ijms, 2023, doi:10.3390/ijms241411725_

Round 1

Reviewer 1 Report

In the present work, Kostyukevich and coworkers administer pure D2O to two mice, causing a dramatic decrease of their well-being (Cit from line 77). After sacrifice, they observe how deuterium has been incorporated into the lipids of the different organs. They do so HPLC-MS/MS.

From the introduction it seems that the paper aims to describe an analytical platform they have developed to observe the lipids where D2O is incorporated once it is administered orally to an animal, so to gain insights into the animals’ lipids metabolism. The novelty, compared with previous works, seems that the observation can be conducted simultaneously in different organs.

Unfortunately, the above aim is my personal educated guess, because the aim of the paper is not stated clearly and cannot be easily inferred by the literature reported in the introduction. Moreover, in the conclusion (line 284) they write that the paper has described “the conventional untargeted lipidomics workflow”. Summarizing, two things are unclear: the aim of the work and whether the work represents a novelty compared to the present literature.

Discussion in not existent

Ethical concerns: In the Institutional Review Board Statement, the authors write that the animal study protocol was approved by the Skoltech ethical comity, but they do not report any protocol number. This seems particularly relevant, because the author have been clear about the sufferance caused to the two animals during the experiment.

Figure 6 and 7 present typewriting mistakes (Crebs instead of krebs). Moreover, it is not clear if they represent previous or novel knowledge.

Several parts can be understood with difficulty, due to poor english

Author Response

In the present work, Kostyukevich and coworkers administer pure D2O to two mice, causing a dramatic decrease of their well-being (Cit from line 77). After sacrifice, they observe how deuterium has been incorporated into the lipids of the different organs. They do so HPLC-MS/MS.

From the introduction it seems that the paper aims to describe an analytical platform they have developed to observe the lipids where D2O is incorporated once it is administered orally to an animal, so to gain insights into the animals’ lipids metabolism. The novelty, compared with previous works, seems that the observation can be conducted simultaneously in different organs.

Unfortunately, the above aim is my personal educated guess, because the aim of the paper is not stated clearly and cannot be easily inferred by the literature reported in the introduction. Moreover, in the conclusion (line 284) they write that the paper has described “the conventional untargeted lipidomics workflow”. Summarizing, two things are unclear: the aim of the work and whether the work represents a novelty compared to the present literature.

Answer: Dear reviewer, thank you very much for the comments. The manuscript was modified in order to better demonstrate the aim and the results. We have performed additional data processing and added new Figure 3 showing the average deuterium distribution for different lipid classes for different organs. This figure allowed to estimate and compare the rate of lipid synthesis in different organs.

We have corrected typos and improved quality of English language.  

Regarding the novelty, to our knowledge HPLC-MS/MS based lipidomics was never applied for the investigation of the in vivo deuterated lipids, moreover, the CID fragmentation spectra of in vivo deuterated lipids were never reported.

Please find the modified manuscript and the file showing the track changes attached.

Discussion in not existent

Ethical concerns: In the Institutional Review Board Statement, the authors write that the animal study protocol was approved by the Skoltech ethical comity, but they do not report any protocol number. This seems particularly relevant, because the author have been clear about the sufferance caused to the two animals during the experiment.

Figure 6 and 7 present typewriting mistakes (Crebs instead of krebs). Moreover, it is not clear if they represent previous or novel knowledge.

Answer: Figure 6 shows the possible ways of the incorporation of the deuterium into the fatty acids.

The figure 7 shows the possible ways of the incorporation of the deuterium into the choline residues of lipids. Using CID fragmentation were observed that the phosphocholine residues of PC lipids do not carry deuterium.

Typewriting mistakes were fixed, thank you.

Reviewer 2 Report

Article Title: Untargeted lipidomics after D2O administration reveals the 2 turnover rate for individual lipids

In this publications the authors have aimed to access the deuterium uptake in the mice biological metabolic pathways, by feeding 100% labelled water over a period of week and tried to access the mice biological fluids and various organs across the body. The idea and the aim behind the study is of great importance. Understanding the lipid turnover and the regulation of the metabolic pathways are highly importance in the development of many diseases. Here researchers have attempted to screen broad range of lipids in an untargeted manner. These has not been attempted these years due to various complexities in the identification and quantification of the isotopically labelled compounds. recently the advancement in the super high resolution instruments are helping to shed some light in this area.

Saying that there are several concerns in this manuscript that needs to be addressed carefully before this article is considered to be publishable.

1. Full article needs to rewritten focusing on the aim and explaining the significance of the results. through out the article there are repetitive sentences, grammatical errors, typo errors and some sentences doesn't make sense. while reading the article i kept loosing interest as the story was not flowing through.

2. As researchers clearly mention that 100% labelled water feeding is toxic to the animal with in a short amount of time, i was wondering if they are measuring the death responses, which is not quite relevant to the aim that they are aiming to do. there is no clarity of why they decided to feed in 100%.

3. line 20: typo error

4. line 80, 81: rewrite the sentence

5. line 92 to 98: rewrite the paragraph. 

6. Sentence "Deuterium uptake is low for brain, heart" seems to be repeated across the article several times.

7. line 99 to 106: paragraph doesn't make sense, re-write

8. Line 112 and 113: not sure why authors selected only 13 lipids. saying that this figure can go to supplimentary as its not very clear and the significance of selecting 13 specific lipids is not mentioned.

9. line 114: i don't understand this sentence. "It is worth noting that the lipid PG 16:0_18:2 demonstrated a considerable deuterium 114 uptake in lung". the relevance of this specific lipid is not clear.

10. Line 128 and 129: not sure what's the significance of this figure. what does "...." mean?

11. Supplimentaty files can't be opened.

12. typo errors in the methodology sections.

13. limitations are not clear in the discussions.

14. not sure the significance of the figure 7

15. In the results section the percentage of label incorporation and its statistical variations between the replicates is not clear. did authors had 3 replicates of labelled and 3 replicates of non labelled mice??

16. Article doesn't have the summary of the lipids detected.

The research topic and the article is of highly significance, it is best to rewrite the article with a focus on a aim and concluding with biological significance and the methodology limitations, would bake this research very interesting.

Article Title: Untargeted lipidomics after D2O administration reveals the 2 turnover rate for individual lipids

In this publications the authors have aimed to access the deuterium uptake in the mice biological metabolic pathways, by feeding 100% labelled water over a period of week and tried to access the mice biological fluids and various organs across the body. The idea and the aim behind the study is of great importance. Understanding the lipid turnover and the regulation of the metabolic pathways are highly importance in the development of many diseases. Here researchers have attempted to screen broad range of lipids in an untargeted manner. These has not been attempted these years due to various complexities in the identification and quantification of the isotopically labelled compounds. recently the advancement in the super high resolution instruments are helping to shed some light in this area.

Saying that there are several concerns in this manuscript that needs to be addressed carefully before this article is considered to be publishable.

1. Full article needs to rewritten focusing on the aim and explaining the significance of the results. through out the article there are repetitive sentences, grammatical errors, typo errors and some sentences doesn't make sense. while reading the article i kept loosing interest as the story was not flowing through.

2. As researchers clearly mention that 100% labelled water feeding is toxic to the animal with in a short amount of time, i was wondering if they are measuring the death responses, which is not quite relevant to the aim that they are aiming to do. there is no clarity of why they decided to feed in 100%.

3. line 20: typo error

4. line 80, 81: rewrite the sentence

5. line 92 to 98: rewrite the paragraph. 

6. Sentence "Deuterium uptake is low for brain, heart" seems to be repeated across the article several times.

7. line 99 to 106: paragraph doesn't make sense, re-write

8. Line 112 and 113: not sure why authors selected only 13 lipids. saying that this figure can go to supplimentary as its not very clear and the significance of selecting 13 specific lipids is not mentioned.

9. line 114: i don't understand this sentence. "It is worth noting that the lipid PG 16:0_18:2 demonstrated a considerable deuterium 114 uptake in lung". the relevance of this specific lipid is not clear.

10. Line 128 and 129: not sure what's the significance of this figure. what does "...." mean?

11. Supplimentaty files can't be opened.

12. typo errors in the methodology sections.

13. limitations are not clear in the discussions.

14. not sure the significance of the figure 7

15. In the results section the percentage of label incorporation and its statistical variations between the replicates is not clear. did authors had 3 replicates of labelled and 3 replicates of non labelled mice??

16. Article doesn't have the summary of the lipids detected.

The research topic and the article is of highly significance, it is best to rewrite the article with a focus on a aim and concluding with biological significance and the methodology limitations, would bake this research very interesting.

Author Response

Article Title: Untargeted lipidomics after D2O administration reveals the 2 turnover rate for individual lipids

In this publications the authors have aimed to access the deuterium uptake in the mice biological metabolic pathways, by feeding 100% labelled water over a period of week and tried to access the mice biological fluids and various organs across the body. The idea and the aim behind the study is of great importance. Understanding the lipid turnover and the regulation of the metabolic pathways are highly importance in the development of many diseases. Here researchers have attempted to screen broad range of lipids in an untargeted manner. These has not been attempted these years due to various complexities in the identification and quantification of the isotopically labelled compounds. recently the advancement in the super high resolution instruments are helping to shed some light in this area.

Saying that there are several concerns in this manuscript that needs to be addressed carefully before this article is considered to be publishable.

Answer: Dear reviewer, thank you very much for the comments. The manuscript was modified in order to better demonstrate the aim and the results. We have performed additional data processing and added new Figure 3 showing the average deuterium distribution for different lipid classes for different organs. This figure allowed to estimate and compare the rate of lipid synthesis in different organs.

We have corrected typos and improved quality of English language.  

Please find the modified manuscript and the file showing the track changes attached.

  1. Full article needs to rewritten focusing on the aim and explaining the significance of the results. through out the article there are repetitive sentences, grammatical errors, typo errors and some sentences doesn't make sense. while reading the article i kept loosing interest as the story was not flowing through.

Answer: We agree. The article was improved. 

  1. As researchers clearly mention that 100% labelled water feeding is toxic to the animal with in a short amount of time, i was wondering if they are measuring the death responses, which is not quite relevant to the aim that they are aiming to do. there is no clarity of why they decided to feed in 100%.

Answer: We decided to feed 100 % in order to achieve the maximum possible deuteration level. Our aim was to observe the deuterium incorporation in lipids using conventional lipidomics workflow.

  1. line 20: typo error

Answer: fixed, thank you!

  1. line 80, 81: rewrite the sentence

Answer: fixed, thank you!

  1. line 92 to 98: rewrite the paragraph. 

Answer: The paragraph was rewritten.

  1. Sentence "Deuterium uptake is low for brain, heart" seems to be repeated across the article several times.

Answer: fixed, thank you!

  1. line 99 to 106: paragraph doesn't make sense, re-write

Answer: The paragraph was rewritten.

  1. Line 112 and 113: not sure why authors selected only 13 lipids. saying that this figure can go to supplimentary as its not very clear and the significance of selecting 13 specific lipids is not mentioned.

Answer: This figure is very important, because here we demonstrate initial experimental results namely the elongated to the right deuterium distribution. We selected those lipids for which we observed high quality spectra in all organs.

The corresponding changes were made to the text.

  1. line 114: i don't understand this sentence. "It is worth noting that the lipid PG 16:0_18:2 demonstrated a considerable deuterium 114 uptake in lung". the relevance of this specific lipid is not clear.

Answer: We agree. The corresponding changes were made to the text.

  1. Line 128 and 129: not sure what's the significance of this figure. what does "...." mean?

Answer: This figure shows the fragmentation pathways of lipids during CID fragmentation. It is important because we are using this mechanisms to analyse the distribution of the deuterium inside the lipids.

  1. Supplimentaty files can't be opened.

Answer: The format of the Supplementary files was changed to .zip.

  1. typo errors in the methodology sections.

Answer: fixed, thank you!

  1. limitations are not clear in the discussions.

Answer: the text was rewritten, thank you!

  1. not sure the significance of the figure 7

Answer: this figure shows the possible ways of the incorporation of the deuterium into the choline residues of lipids. Using CID fragmentation were observed that the phosphocholine residues of PC lipids do not carry deuterium.

 The corresponding changes were made to the text.

  1. In the results section the percentage of label incorporation and its statistical variations between the replicates is not clear. did authors had 3 replicates of labelled and 3 replicates of non labelled mice??

Answer: We performed the experiment for 2 replicates of labeled mice.

  1. Article doesn't have the summary of the lipids detected.

Answer: The table of detected lipids is placed in the Supporting Information.

The research topic and the article is of highly significance, it is best to rewrite the article with a focus on a aim and concluding with biological significance and the methodology limitations, would bake this research very interesting.

 Answer: Thank you very much! We have improved the manuscript to make it suitable for publication.

Round 2

Reviewer 1 Report

The introduction of figure 3 better justifies the publication of the paper. The authors did not have making themselves clear as their main priority, so that understanding the novelty and significance of the work is still a problem.

In order to address my concerns about ethics linked to the suffecence of the animals, the authors just omitted in the new version o the manuscript that the animals suffered. Moreover, in the first version of the paper it was possible to read: "Institutional Review Board Statement: The animal study protocol was approved by the Skoltech ethical comity.". In the new version of the paper the authors just write that "All experiments were carried out in accordance with the ethical principles and regulations recommended by the European Convention for the Protection of Vertebrate Animals used for Experiments".  I leave to the editor the decision about this point.

Readability is now sufficient. 

Author Response

Dear reviewer, we have re-checked the state of the health of animals at the moment of sacrifice with our partner who was responsible for the experiment. We were assured that the animals didn't suffer and that all experiments were performed in accordance with the conventional ethical principles and regulations.

Also, we have re-checked the English and made minor changes to improve the quality of the manuscript.

Reviewer 2 Report

I still think it would be worth to check the english. The story doesn't flow in various sections. except that its good to publish.

line 40: no sure what is "30-th"

I still think it would be worth to check the english. The story doesn't flow in various sections. except that its good to publish

Author Response

I still think it would be worth to check the english. The story doesn't flow in various sections. except that it's good to publish.
Answer: The english was checked once again.

line 40: no sure what is "30-th"
Answer: The text replaced for "1930s", thank, you!